# Psychogastroenterology: A Cure, Band-Aid, or Prevention?

**DOI:** 10.3390/children7090121

**Published:** 2020-09-03

**Authors:** Miranda A. L. van Tilburg

**Affiliations:** 1Department of Clinical Research, College of Pharmacy & Health Sciences, Campbell University, Buies Creek, NC 27506, USA; vantilburg@campbell.edu; Tel.: +1-(910)-814-4913; Fax: +1-(910)-814-5565; 2Department of Medicine, Division of Gastroenterology and Hepatology, University of North Carolina, Chapel Hill, NC 27514, USA; 3School of Social Work, University of Washington, Seattle, WA 98105, USA

**Keywords:** psychogastroenterology, inflammatory bowel disease, functional gastrointestinal disorders, functional abdominal pain, mental health, biomedical model, biopsychosocial model

## Abstract

Psychogastroenterology is a field that focuses on the brain–gut connection. Many children with gut disorders also struggle with psychological and social factors that affect their disease outcomes. Psychological factors have been suggested to be a cure, a band-aid, or a prevention. This article examines the underlying models of disease and health that determine how we understand and treat psychosocial factors in gut diseases. The biomedical and biopsychosocial models are presented and applied to pediatric gut disorders. This article should familiarize clinicians as well as children and their families to the challenges and opportunities for addressing psychosocial factors in gut disease. Psychogastroenterology is best thought of as a cog in a complex treatment machine.

## 1. Introduction

The brain–gut connection has long been known. We are all intimately familiar with the butterflies in our stomachs when we are in love, or the run to the bathroom when we are stressed. Our emotions clearly influence our gut, and evidence is now available that our guts can also influence our mind [1]. Recent studies have shown, for example, that gut microbiota can influence the brain and induce changes in behavior as well as mood [2]. In some gut disorders, such as Irritable Bowel Syndrome, disordered brain–gut axis has been shown to be the main cause of symptoms [3]. In others, such as Inflammatory Bowel Disease, the evidence for the role of psychological factors in the course of the disease and adjustment to the disease has been growing [4]. Increased awareness about the importance of the brain–gut interaction in gastrointestinal disorders has given rise to the field of Psychogastroenterology. Psychogastroenterology focuses on how psychosocial factors play a role in gut diseases. Psychogastroenterologists are clinicians, such as psychologists, psychiatrists, and social workers, who work in integrative or multidisciplinary care of patients with gastrointestinal disorders. The role of this field has been ranging from suggested prevention, a cure, to a Band-Aid for gut disorders. In this article, I will examine the underlying assumptions that drive these expectations. 

## 2. Gut Symptoms Are Either in Your Gut or Your Mind

A common scenario in clinics is the following. A child comes in with a new or worsened gut symptom. Let’s say abdominal pain. A medical workup happens, and all tests appear to be normal. The family is told “nothing is wrong” and sent home. If the child returns with repeated complaints, more tests will be ordered even if the physician does not feel these are warranted. Most likely these tests are normal and quickly the clinician tires of the “demanding” family who seeks a cause and treatment for their child’s symptoms. By now, this child and/or caregiver appears stressed and anxious so is sent to a psychologist for treatment. Let’s unpack the assumptions that underlie this common scenario.

The physician has been looking for what may be biochemically and structurally wrong with the gut causing pain. Repeated requests from the family for symptom relief become frustrating when no biological cause can be found [5]. Since the physician cannot find any biological reason for the pain, and the family appears upset, the pain must be a direct result of the family’s distress. On the other hand, the family is visiting the physician over and over to find a cure and treatment for what is biologically wrong with their child. Although many families acknowledge the role of stress in their child’s pain [6], most still look for a biological explanation of the pain. After all, a child cannot be in this much pain without a clear biological reason. Parent dissatisfaction with the physician is higher if no diagnosis based on biology is given [7]. Parents, whose children are undergoing medical testing for abdominal pain, have indicated the best outcome is finding something is wrong with their child, while the worst outcome is finding their child is healthy [8]. The latter implicates the child’s symptoms are not believed or “must be all in the child’s head”. Thus, both the physician and family agree that a lack of biological explanation of symptoms must mean the cause of the symptoms is psychological. Where the physician is convinced the cause is psychological based on medical tests, the family is not and is worried the physician has missed something. They will look for as second, third, or fourth opinion to find out what is really wrong with their child. One study found that for children with such pain, extensive medical testing is common [9]. Testing can be uncomfortable for the child and is costly but rarely yields any useful information [9].

The underlying model in the above case is the biomedical model. The biomedical model defines health as the absence of disease. A disease can be in the body or the mind. If no biological reason for the symptoms can be found, this must mean the symptoms are caused by a disease of the mind. In the case above, there was no biological cause. The pain must mean the child is anxious or stressed so they are referred to a psychologist for treatment. Physicians diagnose and treat disease, mostly biological disease. Mental health is left to a small number of health care providers such as psychiatrists and psychologists. Stigma against mental health disease is large in both children and their caregivers, which leads to avoidance of diagnosis and treatment [10]. Families who are told their child’s gut symptoms are due to a mental health disorder will often refuse this diagnosis. Note that the biomedical model allows that disease and illness are not the same. Patient’s experiences, social circumstances, or psychological state can influence illness behaviors such as health care seeking, disability, or quality of life. Illness is seen as outside of medicine and asking a physician to reduce illness is unreasonable [11]. Physicians diagnose and treat the (biological) disease. 

The biomedical model is not supported by evidence. Few studies find a direct link between the severity of the symptoms and the disease. For example, we found in a sample of children with inflammatory bowel disease that the amount of gut inflammation is not associated with the severity of symptoms [12]. Inflammatory Bowel Disease is an interesting disease model as patients wax and wane between flares (gut inflammation) and remission (no gut inflammation). While many children become symptomatic as their disease flares, some children are asymptomatic while in a disease flare, while others report symptoms during remission [13]. The biomedical model explains this as follows [14]. Those who are in remission (no disease) and show no symptoms are healthy. Those who are in a flare and symptomatic have a disease. Their symptoms are a rightful complaint and elicit understanding and help. Children who report no symptoms in a flare are stoic. They go on with life despite their disease. We admire these patients but do not assume they are truly asymptomatic. Children who report symptoms are assumed to make the symptoms up or have a mental disorder. These patients are often thought to be demanding and dismissed by physicians [15,16]. Families carry theses same beliefs. No wonder they do not accept the diagnosis that their child is in the latter category. They are not demanding, their child’s symptoms are real, and the physician should keep looking for a biological cause. To get out of this conundrum we need to accept and communicate a different model of health and disease.

## 3. Symptoms Are Due to Bi-Directional Interaction of the Mind and the Body

Through many years of research, we now understand that the biomedical model has major limitations and does not reflect the reality of disease and illness. In 1965, the understanding of pain was expelled forward by the Pain Gate Theory [17]. This posited that pain signals from the periphery to the brain can be disrupted in the spinal cord leading to blunting of pain reports. This gate could be opened or closed by many different factors both biological (e.g., rubbing the painful site) as well as psychological (e.g., mood). Here was a theory explaining why a similar biological cause (kick to the shin) did not have to lead to a similar severity of symptom (pain). The pain gate theory has been improved upon, but its premise has stood the test of time [18]. It opened the door to allow for many non-biological approaches to treating pain. 

About 10 years later, another innovative model changed the landscape of medicine and understanding of health and disease. The biopsychosocial model [19] rejected the separation of body and mind. It stressed that health and disease should be explained from an interaction between biological, psychological, and social factors. The biopsychosocial model is a holistic model and does not reduce our health to only biological factors. It recognizes that all disease is always an interplay between biological and psychosocial factors. These simply do not exist without each other. The biopsychosocial model can explain many phenomena that were unexplained in the biomedical model and hence was quickly accepted. For example, the World Health Organization defined health as: “a state of complete physical, mental and social well-being and not merely the absence of disease or infirmity”. Despite the popularity of the biopsychosocial model in science, the biomedical model remains the prevailing model in our Western culture, which leads to problems such as those discussed in the case above, though this is slowly changing.

In medicine, the gut has always been seen as a special organ. The gut is highly innervated and has been called the “second brain”. The second brain (enteric nervous system) is in direct and constant interaction with our primary brain (the central nervous system). In the 19th century, the brain–gut axis was thought to explain many diseases and social phenomena [20]. Bidirectional associations were proposed with emotional states affecting gut health, and the gut affecting emotions. For example, excessive tea drinking was thought to cause emotional break-down of women [20]. Even as clinical and scientific focus became increasingly reductionist by focusing on biology alone, the brain–gut axis was never completely ignored in gastroenterology. Within this fertile ground and with the increasing acceptance of the gate-control theory as well as the biopsychosocial model, psychogastroenterology found fertile ground to grow as a science and clinical approach to treat gut disorders.

## 4. Psychogastroenterology: The Brain–Gut Axis

Initial psychogastroenterology studies focused on the role of stress as well as anxiety and depression in children with gut disorders [1,21]. Physicians with limited training in psychology would observe their patients as anxious, fearful, stressed, or depressed. The role of psychological distress was most commonly explored in children who had biologically unexplained gut symptoms. Psychological treatments focused on reducing anxiety and stress in these children and met with varied success [22,23,24]. For example, one study found that relaxation alone did not fare as well as relaxation in combination with pain specific guided imagery [25]. As the interest in the brain–gut axis grew, as well as the sophistication of techniques and models, the field exploded. There is now a large evidence base of the association between psychological factors and gut sensory, motor, and immune functioning [1]. Psychological factors can influence the gut through the hypothalamic–pituitary–adrenal axis and central nervous system modulation of gut stimuli. Until recently, the pathways by which the gut influences the mind were not as well known. The gut microbiota-inflammatory pathway is currently the guiding theoretical model of how the gut can influence the brain [26].

Psychological research has identified specific gut cognitions and behaviors such as catastrophizing (fearing the worst and being unable to change this outcome) and somatization (reporting multiple unexplained symptoms) that are associated with exacerbation of gut symptoms [21]. In addition, parents (the primary social environment for the child) are important drivers of disability. When parents worry about the meaning of the symptoms (increased disease threat), they are more likely to visit a doctor, keep their kids home from school, and discourage normal activity [27]. These are factors that can be addressed by treatment. Cognitive behavioral treatment and hypnosis addressing gut specific cognitions, emotions, and behaviors—rather than stress, anxiety, and depression alone—are effective in reducing symptoms and disability in children with gut disorders [28,29,30,31]. Most of this evidence refers to gut symptoms that were initially thought to be biologically unexplained and became labeled as functional gastrointestinal symptoms (the healthy gut functions differently) and now are referred to as Disorders of Gut–Brain Interaction [3,32]. However, increasing evidence is available from other gut disorders as well, such as inflammatory bowel disease.

Even with the wide evidence base on the bi-directionality of the brain–gut axis, the question often arises: Which came first: the brain or the gut? Paying homage to the biomedical model, it is often assumed that psychological difficulties are a cause of functional gut disorders [1] and are co-morbid with organic gut disorders such as inflammatory bowel disease [33]. Co-morbidity assumes that two diseases are present—gut disorder and psychiatric disorder—which are largely independent of each other and require independent treatments. However, the evidence does not support this. For example, in functional abdominal pain disorders, studies have found evidence that anxiety may precede as well as follow the pain [21]. In inflammatory bowel disease, immune dysregulation can explain both gut inflammation as well as depression [34]. If we accept the bidirectional nature of the brain and gut axis, it does not matter which came first. We can now intervene at both the level of the gut as well as the level of the brain. For many patients, integrative treatment is needed [35]. 

## 5. Psychogastroenterology: A Cure, Band-Aid, or Prevention?

Given the developments in the field of psychogastroenterology, the need for integrative therapies is becoming widely accepted. Many pediatric gastrointestinal clinics are integrating psychologists into their practices. Practical barriers around reimbursement and lack of trained therapists hamper access to integrated care for many patients [35]. However, this is a development that cannot be stopped, and these services will become more available over time. The question remains how clinicians and patients view the role of psychologists in gut diseases. 

### 5.1. Psychogastroenterology as a Cure

Physicians and psychologists who adhere to the biomedical model will see psychological therapy as separate from medical treatment. The psychological therapy is aimed at treating mental health issues and rarely addresses gut-specific cognitions, emotions, and behaviors. Mental health issues may be either seen as causing the gut symptoms (*cure*) or exist in addition to the gut symptoms. Where mental health is the domain of the psychologist or psychiatrist, physicians treat the gut issues. In this model, patients will feel misunderstood, labeled as crazy, and often do not follow-up with the referral to a psychologist.

### 5.2. Psychogastroenterology as a Band-Aid

The biomedical model also allows that psychological factors may influence illness behaviors. For example, an anxious mother is more likely to take her child to a doctor for nausea. Studies have shown that in the presence of a gut disorder, psychological factors can exacerbate symptoms, increase disability, and decrease quality of life [12,21,36]. In this model, psychological therapy is offered to help children and their families deal with the difficulties of living with a chronic disease so they can live a fuller life. This is far more acceptable to patients. However, it does not directly address gut symptoms themselves, solely the disability of living with the symptoms. 

### 5.3. Psychological Treatment as Prevention

If we assume that some gut disorders, such as functional disorders, are caused by mental health, we can prevent these gut disorders by treating mental health before gut symptoms appear. For example, we can aim to prevent functional abdominal pain by reducing stress in schools [37]. This is currently not a widely adopted approach.

Thus, it seems the biomedical model advocates for psychologists in the treatment of gut disorders (cure/prevention or Band-Aid). Why would we need a biopsychosocial approach? First, the biomedical approach often leans heavily on psychogastroenterology as a cure. As discussed above, this leads to frustration from families and avoidance of treatment. When psychologists focus on a cure, they treat mental health diagnoses such as anxiety or depression instead of gut-specific cognitions, emotions, and behaviors. Although treating clinical anxiety and depression disorders is needed, they do not address the gut complaints. Rather, it has been found that addressing gut-specific cognitions, emotions are the most effective part of psychological treatments for gut disorders [38,39]. Thus, the biomedical approach to child gut disorders separates the treatment of the gut from the treatment of the brain, and this is associated with suboptimal outcomes.

I propose a 4th approach to Psychogastroenterology: Psychological treatment as a cog in a complex treatment machine. The biopsychosocial approach emphasizes psychological treatment as a cog in an integrative treatment approach that addresses both psychosocial as well as biological factors in gut complaints. Research has shown that integrative treatment approaches are effective in pediatric gastrointestinal clinics [40,41]. Integrative treatment synthesizes disciplines into a coordinated, interactive whole. Physicians and psychologists (as well as other clinicians such as dieticians), develop joint treatment plans. Often patients are being seen jointly or in the same practice which reduces stigma and problems with access to treatment. No matter the practical considerations of when and where the child is seen, integrative care has joint treatment plans and teams. 

In contrast to the biomedical approach, where psychological therapy is often not considered until the child has developed clinically significant psychiatric distress or life-altering disability [42], integrative treatment approaches can and should be offered to all children in all stages of a disease. For example, after the diagnosis of a life-altering disease, such as inflammatory bowel disease, support to adjust to new life requirements is helpful for all patients. In addition, integrative care models practice preventative strategies by focusing on resilience [43] in the face of a gut disorder to avoid future maladaptive problems. These preventative approaches are currently underutilized.

Thus, integrative approaches centralize the gut symptoms. The family is reassured that they are not labeled crazy and provided treatment that deals with their primary concern: The child’s gut symptoms. Compared to multidisciplinary medical care offered in many clinics, where patients usually separately and consecutively see a gastroenterologist and psychologist, integrative care, where patients are being see jointly and in parallel, is better at reducing gut symptoms as well as psychological distress, increases patient satisfaction with treatment, reduces barriers such as stigma and access, and reduces health care costs [28,35]. Thus, the biopsychosocial model allows access to integrative care that is focused on the improvement of gut symptoms and associated disability as well as preventative strategies to avoid future exacerbation of gut and mental health.

## 6. Conclusions

In conclusion, all researchers and clinicians should be aware of the model of health and disease underlying their study and treatment assumptions. In the case of pediatric gut disorders, simply measuring anxiety or adding a psychologist to your practice does not make you practice psychogastroenterology. This field takes a more holistic approach, and the evidence base is growing that integrative psychogastroenterology improves patient outcomes. One of the main challenges will be to obtain buy-in from families. Mentioning the role of stress or mental health and suggesting psychological care for a gut disorder communicates that the clinician believes the symptoms are all “in the child’s head”. Clinicians should expect and respect this reaction. These families are not trying to be difficult. Most families will accept psychological care when presented in the right way. This includes reassurance that the symptoms are real and caused by physiological changes in the gut as well as the brain, as well as a thorough explanation of the brain–gut axis, introducing psychological care as a treatment for the gut rather than mental health symptoms, and incorporating psychologists as part of the treatment team. The explanation of gastrointestinal symptoms to families within the framework of the biopsychosocial model is key in their acceptance and treatment success. While children see a mental health provider, it is very important for the doctor to remain available at all times, including scheduling follow-up appointments and giving advice on what level of symptom change should require a visit. Not doing so suggest to the family that their child’s symptoms are still not believed by their doctor. As the field of psychogastroenterology grows, and more integrative care becomes available, the care for children with gut disorders will improve. 

Below are several resources that are helpful for those providers who would like to start integrative psychogastroenterological care in their clinics:Tips for physicians on how to help a family engage with an evidence-based treatment plan for chronic pain as explained by Schechter and colleagues [44].How to talk to patients about chronic pain can be found in Coakley and Schechter [45].Best practice on incorporating psychogastroenterology into the management of digestive disorders is described by Keefer and colleagues [46]. Although this is written for adult providers, much can be applied in pediatric practices as well.Implementing psychological therapies for gastrointestinal disorders in pediatrics by Reed and colleagues [47]. This also contains resources on how to find a psychologist to work with in a gastrointestinal clinic.

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
