# Peer review of "Psychogastroenterology: A Cure, Band-Aid, or Prevention?"

_children, 2020, doi:10.3390/children7090121_

Round 1

Reviewer 1 Report

Thank you for the opportunity to review this overview of psychogastroenterology. The review is well-written and raises a number of important points regarding the difference between biomedical and biopsychosocial models as well as clinical examples. I have one major substantive comment and couple places for minor improvements.

  1. The three options presented, A Cure, Band-Aid, or Prevention, all followed with arguments for or against, but it was unclear which of these three options the author would advocate for as being consistent with the role of psychogastroenterology or if they would argue for a 4th option. Prevention seemed to come closest, but this does not fully capture the role and essence of integrated psychological care for disorders of the brain-gut axis as described further in text. Is there a fourth word or metaphor that would fully capture the essence of the ideal role of psychogastroenterology? I think making a strong case for a fourth option as the ideal role would strengthen the argument and give the reader closure.
  2. In the conclusion, it would be helpful to provide at least a resource or two on how-to describe the role of a psychologist in a gastroenterology practice. Perhaps referencing the pain metaphors article or another resource that clinicians could read if they would like to learn how to discuss the role of psychology would be helpful (http://possiblemind.co.uk/wp-content/uploads/2015/06/Chronic-Pain-Metaphors-for-Children.pdf).
  3. There are a few places where some additional references would be helpful.
    1. On pg. 2 lines 86-96—are there any references to support child dismissal for being asymptomatic or symptomatic?
    2. On pg. 3, lines 135-136, citations for the fact that psychological treatments are met with varied success?

Author Response

We thank the reviewer for their excellent suggestions.

Reviewer 2 Report

This is a well-written review of the change in paradigm and understanding of disorders of the brain-gut interaction over  recent years, specifically focusing on Psychogastroenterology. The author explains the difference between the biomedical and biopsychosocial model of illness and how research has changed our understanding from a dichotomous biomedical model to a bi-directional and integrative understanding of the mind and body, as well as the implications for treatment of these often difficult to treat Disorders of Gut-Brain interaction, formerly known as functional gastrointestinal disorders.

The scenario provided to explain the biomedical model serves well as an illustration of the limitation of the biomedical model in patients with functional gastrointestinal symptoms. Luckily this has been changing and while many patients are still being treated according to the biomedical model, there have been significant changes in our understanding of the limitations of the biomedical model in recent years, which I think is important to state so the reader does not get the impression that all patients are still treated according to the biomedical model. Would recommend adding the following:

Page 3, Line 118: "lead to the problems such as the case discussed above, though this is slowly changing."

Page4, Line 166: Recommend adding a reference that has shown that multidisciplinary care is effective [Beinvogl et al, CGH 2019]

Page 5, line 205: It appears this sentence should read "...biopsychosocial approach does not advocate.."

Page 5, Line 229. As the authors state, explaining the role of stress or mental health and suggesting psychological care for a gut disorder will meet much wider acceptance if the symptoms are explained appropriately. It is essential to acknowledge that the symptoms are real, explaining that there are physiologic changes occurring on the level of the gut and the brain that play a role in the development and maintenance of functional gastrointestinal symptoms and to explain how these can be very positively influenced by psychological treatment as it affects pain modulation. The explanation of functional gastrointestinal symptoms to families within the framework of the biopsychosocial model is key in their acceptance and treatment success.

Few minor edits:

  • Second author name is missing (and after the first author implies there is a second author)
  • Page 1 Line 43, delete 'the'
  • Page 5, line 205 should read "the bioppsychosocial approach does not advocate waiting for"

Author Response

Thanks for the great suggestions
